# Educational cooperation in the perspective of tripartite evolutionary game among government, enterprises and universities

**Shuangzhi Zhang** *

College of Teachers, Chengdu University, Chengdu, China

\* zhangshuangzhi@cdu.edu.cn

## Abstract

Government-enterprise-university synergy (GEUS) is an effective way to mobilize government, enterprises, and universities to collaborate on education, but these three parties involved in GEUS may, out of bounded rationality, choose to collaborate in ways that benefit themselves and harm others. To guide the three parties to better cooperation, this study creates an evolutionary game model among the three parties and evaluates the applicability and validity of the model by selecting the educational cooperation data in Beijing. It is shown that participation in education cooperation is the best course of action for all three parties. The intensity of willingness to participate in the GEUS is on the order of high to low for universities, enterprises, and the government. If the three parties wish to accomplish education collaboration sooner, they can increase default payments, boost government revenues, raise corporate participation in distribution, and reduce government and government spending. These results highlight the inherent regularities of GEUS and provide concrete implementation strategies to improve the efficiency of education cooperation.

## 1. Introduction

University graduates' employment is a global issue that impedes social and economic growth [1]. Cooperation between enterprises and universities is critical to resolving the issue of graduate employment [2]. Additionally, it simplifies the process of incorporating educational technology into actual teaching [3]. However, to achieve such cooperation, the government must occasionally actively promote it. As a type of "1+1+1" cooperative education, Government-Enterprise-University Synergy (GEUS) integrates the advantages of the government, enterprises, and universities in the social division of labor and resource sharing, achieving not only the docking of talent cultivation and employment but also the coupling of knowledge production and knowledge commercialization [4]. However, the limited rationality of the participants in GEUS leads to inconsistencies in their perceptions of the benefits [5]. Because of their short-sightedness, the participants may not choose GEUS and so miss out on the value-added benefits provided by the synergy. Therefore, how to promote GEUS becomes a crucial research topic in the market economy by leveraging the respective advantages of government, enterprises, and universities.

**Data Availability Statement:** All relevant data are within the paper and its Supporting Information files. The data in this paper comes from China Business Intelligence Website. The authors have extracted the parameter values about government,

universities and enterprises from the report of Innovation and Entrepreneurship and University-Industry-Research 2024-2029. As this report is a paid report, the authors paid a fee and are not able to share the report data. They have included the link to this report in the text as a reference and listed the specific data used in this article in Table 6 (please refer to reference [56] and Table 6.). The research data extracted from this report of Innovation and Entrepreneurship and University-Industry-Research 2024-2029 is shown in Table 6.

**Funding:** This work was supported by The MOE (Ministry of Education in China) Project of Humanities and Social Sciences (Grant No. 22YJC630207); Chengdu Soft Science Research Project (Grant No. 2023-RK00-00019-ZF); Chengdu Philosophy and Social Sciences Research Project (Grant No. NY2320231279). The funders had no role in study design, data collection and analysis, decision to publish, or preparation of the manuscript.

GEUS is a modern education reform model that takes enterprises' demand as the guide, univ GEUS is a modern education reform model that takes enterprises' demand as the guide, university talents as the basis, and the government integrates universities and enterprises to promote talent training and scientific research transformation. It fundamentally resolves the issue of the discrepancy between university talent education and social talent requirements by linking enterprise production, which concentrates on gaining practical experience, with university education, which focuses on knowledge transfer [6]. In other words, it completes the transformation of classroom knowledge into technical capital [7].

However, because the government, enterprises, and universities don't know what kind of decision each other will make when deciding whether or not to engage in educational cooperation, it's difficult to synchronize decision-making among the three, making it hard to promote educational cooperation smoothly [8]. The conduct of numerous parties involved in decision-making is perfectly rational in the classical game, and everyone makes well-considered decisions with completely open information [9]. However, because of the three cross-organizational levels and barriers, the condition of complete information does not apply to GEUS. Therefore, cooperation among government, enterprises, and universities is a constrained rational decision made by three parties under information asymmetry, in which the three parties gradually learn from each other's decisions and improve their own strategies based on each other's decisions. The three parties eventually developed a stable cooperative intention to maximize their own interests through continual modification in the conflict resolution process. As a result, the evolutionary game method should be used to investigate government-business-academic collaboration.

Currently, most of the existing studies on educational cooperation discuss only the interest game between enterprises and universities, ignoring the leading role of the government [10–12]. The government is more often considered as an exogenous variable in educational cooperation rather than being included as a member of the game involved [13]. This disconnect ignores the co-constructive role of government in education cooperation and fails to recognize that cooperation between government, enterprises, and universities is a synergistic and symbiotic system of government, enterprises, and colleges and universities.

To address the shortcomings of previous studies, this paper aims to incorporate the symbiotic nature of government into the tripartite evolutionary game system, viewing government as a co-constructor of educational cooperation rather than merely a facilitator. By constructing a tripartite evolutionary game model of education cooperation that includes the government in the symbiotic system, this paper not only identifies the optimal game strategies of the government, enterprises, and universities in education cooperation, but also provides concrete implementation suggestions to promote the participation of the three parties in education cooperation. Overall, the conclusions of this paper make up for the lack of game strategy research on the role of the government in educational cooperation in academia and provide a concrete program to strengthen the synergy between the government, enterprises and universities in practice.

The key to GEUS is to engage participants in cooperative education consciously and proactively. Although choosing synergy by all three parties is the best participation strategy, individuals may make irrational decisions to not synergize due to limited rationality [14]. For example, the government may fear that enterprises or universities are not willing to participate and that their impulsive leadership choices may result in a lack of return on their initial investment in GEUS. Enterprises may suspect that the synergy's revenue will fall short of expectations and that they won't be able to balance their income and expenses [15] Universities are afraid that they will suddenly withdraw from the collaboration and invest time, energy, and money in talent training for nothing [6]. Such skeptical attitudes will not only hinder synergy

but will also lead to its collapse [16]. Therefore, it is necessary to explore how to control such oscillations with the aid of a three-way evolutionary game model. The research questions in this paper are threefold:

1. Is there a stable evolutionary strategy in GEUS?

2. What kind of interactions exist in the decision-making of the three parties?

3. How to urge the three parties to choose GEUS better?

To answer the above questions, this paper constructs an evolutionary game model of GEUS and verifies the validity of the model through theoretical analysis and simulation analysis. Furthermore, the conclusions reached in this research fully address the problems raised above. First, there are stable evolutionary strategies (1, 1, 1) in GEUS, in which the government, enterprises, and universities all choose to participate. The simulation of the evolution of GEUS in Beijing confirms that the evolutionary trajectory of GEUS gradually evolves from an unstable strategy to a stable strategy. Second, university participation decisions are not influenced by government or enterprise. However, the participation decisions of enterprises and governments are susceptible to change. Negative enterprise participation can cause the government to adopt a non-participation strategy, and negative government participation can cause enterprises to adopt a wait-and-see attitude. Third, the evolution of a stabilization strategy can be accelerated by increasing default payments, boosting the percentage of benefits allocated to enterprises, expanding government revenues, cutting government expenditures, and slashing enterprise expenditures. The general framework of the study is shown in Fig 1.

The remainder of this article is organized as follows: the second section examines the literature on GEUS and evolutionary games. In the third section, the GEUS evolutionary three-

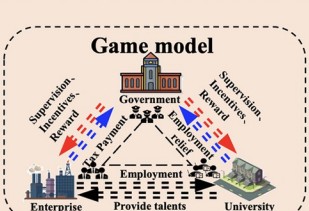
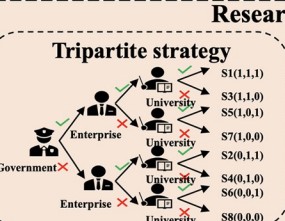
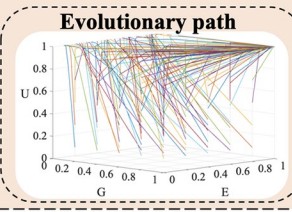

**Research method**

**Game model**

**Tripartite strategy**

S1(1,1,1)
S3(1,1,0)
S5(1,0,1)
S7(1,0,0)
S2(0,1,1)
S4(0,1,0)
S6(0,0,1)
S8(0,0,0)

**Simulation parameters**

Initial participation willingness

Default amount

Benefit distribution ratio

Incentive allocation ratio

Income and expenditure

**Evolutionary path**

**Rational explanation**

**Tripartite Participation**

$G2j - G1k < 0$

$-P + E2f - G1ku < 0$

$-Q + gU2 - G1jv < 0$

**Enterprise Involvement**

$-G2j < 0$

$Q - gU2 < 0$

$-Q + E2f < 0$

**University participation**

$-G2j < 0$

$-P + gU2 < 0$

$P - E2f < 0$

**Non-participation**

$-G2j < 0$

$P - gU2 < 0$

$Q - E2f < 0$

**Research result**

**Participation willingness**

The government will be well-positioned to act as a motivator and arbitrator when both universities and enterprises like to participate.

**Default amount**

Higher default fees should be imposed to expedite the GEUS process, too low default fees will eliminate the incentive for enterprises and governments to participate.

**Benefit and Incentive ratio**

GEUS can indeed be promoted by adjusting the benefits allocation ratio, and the larger the enterprise's share in the allocation ratio, the faster the three parties will choose GEUS.

**Income and expenditure**

Increasing government revenues while decreasing government and enterprise expenditure can hasten GEUS, whereas cutting university expenditure has little influence on GEUS.

**Fig 1. Research framework.**

party game model is developed. The fourth section presents the theoretical and simulation analyses of the model. The results are discussed in the fifth section. The sixth section outlines the contributions and limitations of the research.

## 2. Related works

### 2.1 Government-enterprise-university synergy

Cooperative education originated in 1906 at the University of Cincinnati in the U.S [17]. The establishment of the World Association for Cooperative Education in 1983 marked the trend of educational reform that relied on collaborative and integrated education [18]. Cooperative education cuts the cost of education and increases the competitiveness of graduates [19]. Subsequently, MIT and British universities have adopted this integrated and practical form of education for educational reform [20]. The practice has proven that enterprise-university synergy is an internationally recognized way to cultivate innovative talents [4, 21]. Academic research in this area is growing annually due to the considerable societal advantages [4, 22].

Currently, academic research on GEUS concentrates on innovation systems. Scholars generally agree that GEUS can improve innovation in universities and enterprises [23] and facilitate the transformation of intellectual property [24]. GEUS provides a driving force for knowledge integration and innovation [25]. Through the application, it can introduce cutting-edge technology to universities and offer paramount theories for enterprises to overcome bottlenecks [26]. In fact, GEUS can also facilitate cooperative education, a topic that scholars have long neglected to study with GEUS [27].

In GEUS research, only case interviews and econometric analysis methods are used. Econometric analysis is employed in 32% of studies, while case study data is applied in approximately 56% of research [22]. For instance, Arshed et al. used an econometric survey to study GEUS in 139 countries between 2007 and 2018 [23]. Tang et al. investigated the relationship between GEUS regional proximity and product innovation performance in 166 universities in Guangdong [28]. Chen et al. explored the content of collaboration in GEUS using a bibliometric approach [4]. Akhtar et al. adopted a quantitative approach based on network analysis theory and techniques to investigate the interaction between participants in GEUS [25]. Yu & Yuizono discussed the impact of heterogeneity in geographical location on GEUS based on a hierarchical regression approach [29].

Notably, GEUS is a multi-participant decision system, where the participants' strategies are ever-changing in a complex and variable game [30]. However, few studies use game theory to explore how to promote GEUS [16]. Let alone that some of them still follow a static game approach where decisions are made at once [31]. In contrast, the real GEUS is a dynamic development process of decision changes [32]. Through constant trial and error, the participants in GEUS learn and develop. Eventually, they will select a stable approach by adapting and adjusting their strategies [33]. Therefore, employing evolutionary game theory to analyze GEUS is beneficial.

### 2.2 Evolutionary gaming

Game theory is a systematic science that studies the optimal strategy of multiple participants in the gain or loss of interests [34]. Evolutionary game theory is a game theory that takes biological evolution and finite rationality as the theoretical basis [35] and combines traditional game theory and dynamic evolutionary processes [36].

The traditional game theory assumes that the game subject is perfectly rational and the game environment is transparent with complete information [37]. But these conditions are often too harsh to meet the actual situation [38]. Nash presented an evolutionary game theory

based on finite rationality and imperfect knowledge to be more accurate. He points out that the game subject must learn, change, and optimize its game strategy over a lengthy period to get the game system to attain a stable equilibrium state. Only after several iterations of the game can the players reach equilibrium [39].

According to evolutionary game theory, each participant adopts the maximization of their interests and repeats the game while continuously adjusting their behavioral strategies with limited rationality and incomplete information [40]. Compared to static games where the ultimate choice is made just once, evolutionary games are more realistic. The equilibrium point of an evolutionary game is not a singular entity, and the game system may simultaneously contain numerous evolutionary stable strategies depending on the scenario and variables [41].

In the game of cooperative education, a few scholars have begun to apply evolutionary game models between universities and enterprises [42]. For example, L. Chen et al. observed that corporate incentives have a favorable impact on collaboration and innovation between enterprises and universities [43]. Some other scholars have introduced government policies into the GEUS, expanding the research boundary of the game [16]. According to Yang et al., reasonable government incentives and default payments can encourage synergy. Appropriate tax rates and penalties can create synergy within the regulatory system [24]. In government-business science, Zhou & Luo discovered that the decisions of the three sides are mutually influential, and profit and cost are the primary factors that influence whether the game system can converge [44]. Although the incorporation of government into the evolutionary game of GEUS is a breakthrough, they only evaluate the impact of government penalties and incentives on GEUS and neglect to consider the effect of government benefits on evolution in GEUS.

## 2.3 Literature summary

As mentioned above, the majority of extant GEUS studies focus on the field of collaborative innovation while ignoring the impact of GEUS on cooperative education. In addition, most GEUS research methodologies have concentrated on interview analysis and statistical analysis, with the usage of evolutionary simulation being quite rare. GEUS is a symbiotic ecosystem composed of government, enterprises, and universities. Although some scholars have adopted static game theory to explore the stability decisions of the three parties in GEUS, the GEUS in the real situation is a dynamic process in which the participants are learning and exploring time and again to find the stability strategies. Evolutionary games provide a way to study the dynamic game process of GEUS. Furthermore, existing studies do not adequately consider the government's symbiotic role in the dynamic game process of GEUS. As a result, this research develops a symbiotic evolutionary model of GEUS for cooperative education and devotes it to evolutionary simulation to address how to motivate the participants of educational GEUS.

## 3. The GEUS evolutionary game model

### 3.1 The parameter system of the GEUS model

The mindset of avoiding harm makes the three parties in GEUS unable to make rational decisions when faced with value implications, and information asymmetry leads to randomness in each party's decision. To further analyze what decisions each party will make in their interest boundaries, I make the following assumptions:

Hypothesis 1: In the presence of limited rationality and information asymmetry, each of the three participants has two strategy options (participate, not participate). The likelihood of government participation is ($x$), while the likelihood of non-participation is ($1-x$). The probability of enterprise participation is ($y$), and the probability of non-participation is ($1-y$). The chance of a university participating is ($z$), and the possibility of it not participating is ($1-z$). The

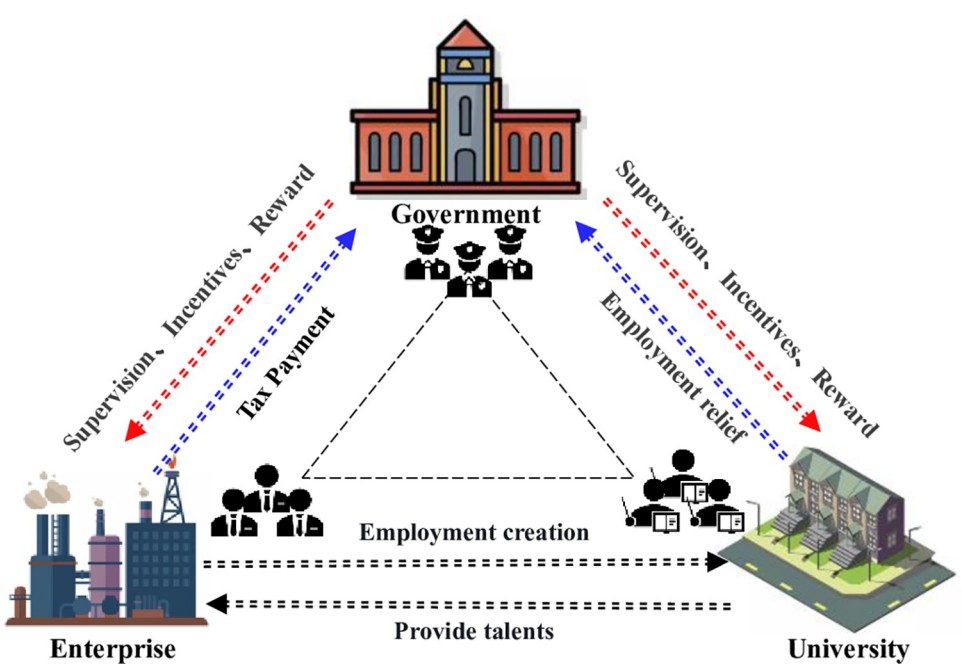

**Fig 2. The inner logic of GEUS.**

government's decision to participate indicates that they want to solve the problem of vicious competition in the labor market caused by the sloppy economic growth method, whereas their decision not to participate indicates that they are concerned that the invested financial support will not change the existing educational structure imbalance. Enterprises choose to join if they want a highly qualified workforce to expedite their company development, and they choose not to participate if they believe the benefits provided by university-trained graduates will outweigh the costs of engaging in the synergy. Universities can opt to engage to show that they desire to enhance the system for cultivating highly skilled talent, or they can choose not to participate to show that they do not believe that talent developed with significant financial investment can achieve exact skill and function matching. Fig 2 depicts the three parties' evolving tactics.

Hypothesis 2: Ideally, all three parties would select to participate, but enterprises and universities may take advantage of opportunistic profits, resulting in a less-than-ideal participation strategy (Liao & Liao, 2022). Because the presence of opportunism might lead to moral hazard, the government, as the dominating player, not only performs the function of symbiosis but also assumes regulatory responsibilities [45]. There is a symbiotic relationship between the government, enterprises, and universities. Consequently, the government's profits through GEUS may be distributed to enterprises and universities, with the proportion of enterprises being ($u$) and the proportion of universities being ($v$). The profits of the government, enterprises, and universities not participating are ($G1$), ($E1$), and ($U1$), respectively; The expenditures when not participating are ($G2$), ($E2$), and ($U2$), respectively; The proportion of additional costs required when participating are ($j$), ($f$) and ($g$), respectively. The percentage of value-added benefits that the government can obtain from GEUS is ($k$). To prevent betrayal between the enterprise and the university, the two need to sign a default damages treaty, which is enforced by government supervision. When the university participates but the enterprise does not, the default amount given by the enterprise to the university is ($P$). In the adversarial

**Table 1. Description of the parameters of the GEUS evolutionary game.**

| Participant | Parameter | Connotation |
|---|---|---|
| Government | $G1$ | The benefits of government non-participation. |
| | $G2$ | The expenditures of government non-participation. |
| | $k$ | Percentage of additional benefits of government participation. |
| | $j$ | Percentage of additional expenditures of government participation. |
| | $u$ | Percentage of government benefits to enterprises after the success of GEUS. |
| | $v$ | Percentage of government benefits to universities after the success of GEUS. |
| | $m$ | The percentage of government incentives given to enterprises when both universities and enterprises do not participate. |
| | $n$ | The percentage of government incentives given to universities when both universities and enterprises do not participate. |
| | $x$ | Probability of government participation. |
| Enterprise | $E1$ | The benefits of enterprise non-participation. |
| | $E2$ | The expenditures of enterprise non-participation. |
| | $P$ | Compensation from the enterprise to the university when the university participates and the enterprise defaults. |
| | $f$ | Additional percentage of expenses when enterprises participate. |
| | $y$ | Probability of enterprises participation. |
| University | $U1$ | The benefits of university non-participation. |
| | $U2$ | The expenditures of university non-participation. |
| | $Q$ | Compensation from the university to the enterprise when the enterprise participates and the university defaults. |
| | $g$ | Additional percentage of expenses when university participate. |
| | $z$ | Probability of university participation. |

case, the default amount given by the university to the enterprise is ($Q$). Normally, $P = Q$. When enterprises and universities are unwilling to engage, the government will allocate a portion of the funds to encourage them to participate in GEUS, where the proportion of enterprises is ($m$) and the proportion of universities is ($n$). The symbols used in this model are shown in Table 1.

Hypothesis 3: According to the deterministic effect of prospect theory and the principle of loss aversion [46], The benefit gained by any of the three parties should be greater than the benefit gained by not participating.

It is worth emphasizing that this paper is inspired by Yin Shi's research on engineering ethics education [47]. In scholar Yin Shi's study on collaborative education, the author emphasizes that cooperative education is a combination of participating objects under certain rules and that this whole achieves educational synergy through resource sharing and optimal allocation. This viewpoint coincides with this paper's promotion of government, enterprises, and universities as a symbiotic system that promotes educational cooperation. Nevertheless, there are some differences between the evolutionary game indicator system constructed in this paper and the implementation effect evaluation system constructed by Yin Shi, who constructed an implementation effect evaluation system with 27 indicators from the perspective of cultivating education, collaborative education, and situational education, while this paper constructs an evolutionary game indicator system with 19 indicators from the perspective of participating members. In other words, Yin Shi's study discusses collaborative education through the subject's participation in scenarios, while this paper studies educational cooperation through the subject's participation in decision-making. In terms of research methodology, this paper is also inspired by Yin Shi's work on three-way games [48]. This author discusses the

**Table 2. Benefit matrix of the three parties in GEUS.**

| Strategies | Government | Enterprise | University |
|---|---|---|---|
| S1(1,1,1) | $G1(1+k)-G2(1+j)$ | $E1+G1ku-E2(1+f)$ | $U1+G1jv-U2(1+g)$ |
| S3(1,1,0) | $G1-G2(1+j)$ | $E1-E2(1+f)+Q$ | $U1-U2-Q$ |
| S5(1,0,1) | $G1-G2(1+j)$ | $E1-E2-P$ | $U1-U2(1+g)+P$ |
| S7(1,0,0) | $G1-G2(1+j)$ | $E1+G2jm-E2$ | $U1+G2jn-U2$ |
| S2(0,1,1) | $G1-G2$ | $E1-E2(1+f)$ | $U1-U2(1+g)$ |
| S4(0,1,0) | $G1-G2$ | $E1-E2(1+f)+Q$ | $U1-U2-Q$ |
| S6(0,0,1) | $G1-G2$ | $E1-E2-P$ | $U1-U2(1+g)+P$ |
| S8(0,0,0) | $G1-G2$ | $E1-E2$ | $U1-U2$ |

game among government, demand firms, and supply firms in technological innovation in the manufacturing industry, which inspires me to explore the game of educational cooperation among government, firms, and universities. In addition, Yin Shi's concept of symbiosis in collaborative innovation systems also provides support for this paper's attempt to construct a symbiosis system between government, enterprises, and universities [49]. However, the objective of this work is educational cooperation in education-oriented employment, which is distinct from the concept of collaborative invention that Yin Shi focuses on in game research.

### 3.2 The reasoning process of the GEUS model

Based on the assumptions stated above, a tripartite-oriented payoff matrix is developed in this paper, as shown in Table 2. The strategy S1(1,1,1) denotes that all three parties have decided to participate, whereas the strategy S8(0,0,0) denotes that all three parties have decided not to participate. The game choice of the three parties is shown in Fig 3.

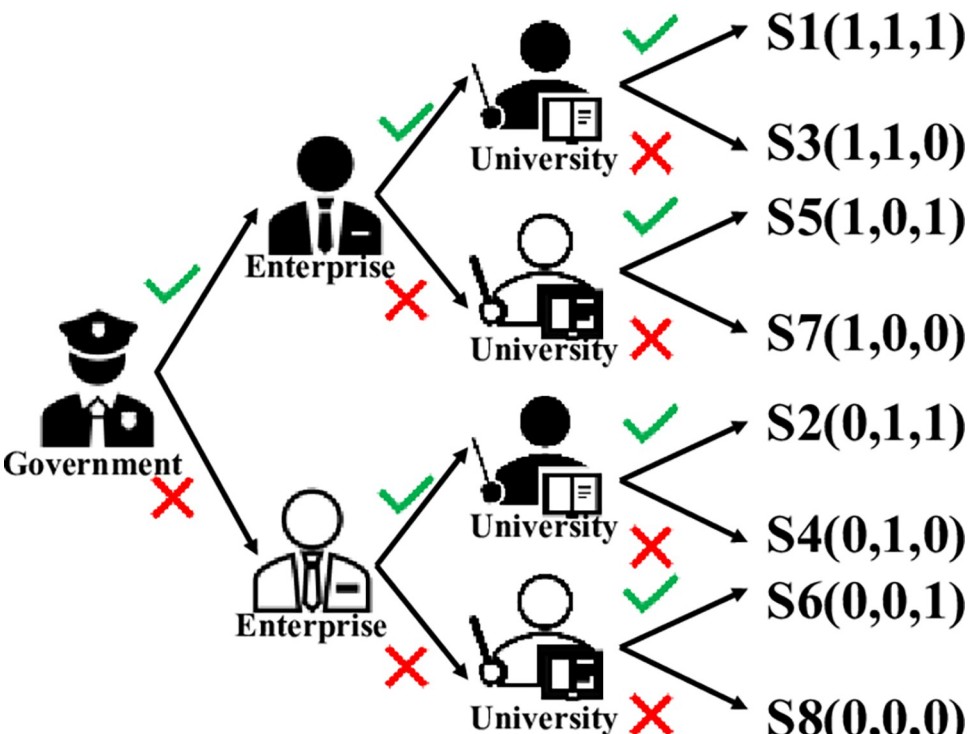

**Fig 3. The selection strategy of the three parties in GEUS.**

The benefits of the parties in Table 2 can be applied to calculate the expected benefits $E_{i1}$ for participation in GEUS, the expected benefits $E_{i2}$ for non-participation in GEUS, and the average expected benefits $\bar{E}_i$ of the three parties in GEUS, ($i = x, y, z$). where $x$ represents the government, $y$ represents the enterprise, and $z$ represents the university.

$$
\begin{aligned}
E_{x1} &= y*z*(G1*(1+k) - G2*(1+j)) + y*(1-z)*(G1 - G2*(1+j)) + (1-y)*z*(G1 \\
&\quad - G2*(1+j)) + (1-y)*(1-z)*(G1 - G2*(1+j)) \\
&= G1 - G2(1+j) + (-G1 + G1(1+k))y\,z
\end{aligned}
\tag{1}
$$

$$
\begin{aligned}
E_{x2} &= y*z*(G1 - G2) + y*(1-z)*(G1 - G2) + (1-y)*z*(G1 - G2) + (1-y)*(1 \\
&\quad - z)*(G1 - G2) \\
&= G1 - G2
\end{aligned}
\tag{2}
$$

$$
\bar{E}_x = x*E_{x1} + (1-x)E_{x2} = G1 - G2(1+jx) + G1\,k\,x\,y\,z
\tag{3}
$$

$$
\begin{aligned}
E_{y1} &= x*z*(E1 + G1*k*u - E2(1+f)) + x*(1-z)*(E1 - E2*(1+f) + Q) + (1 \\
&\quad - x)*z*(E1 - E2*(1+f)) + (1-x)*(1-z)*(E1 - E2*(1+f) + Q) \\
&= E1 + Q - E2(1+f) - Qz + G1\,k\,u\,x\,z
\end{aligned}
\tag{4}
$$

$$
\begin{aligned}
E_{y2} &= x*z*(E1 - E2 - P) + x*(1-z)*(E1 + G2*j*m - E2) + (1-x)*z*(E1 - E2 - P) \\
&\quad + (1-x)*(1-z)*(E1 - E2) \\
&= E1 - E2 - Pz + x(G2\,j\,m - G2\,j\,m\,z)
\end{aligned}
\tag{5}
$$

$$
\begin{aligned}
\bar{E}_y &= y*E_{y1} + (1-y)E_{y2} \\
&= E1 + Qy - E2(1+fy) + G2\,j\,m\,x(-1+y)(-1+z) - Pz + (P - Q + G1\,k\,u\,x)y\,z
\end{aligned}
\tag{6}
$$

$$
\begin{aligned}
E_{z1} &= x*y*(U1 + G1*j*v - U2*(1+g)) + x*(1-y)*(U1 - U2*(1+g) + P) + (1 \\
&\quad - x)*y*(U1 - U2*(1+g)) + (1-x)*(1-y)*(U1 - U2*(1+g) + P) \\
&= P + U1 - (1+g)U2 - Py + G1\,j\,v\,x\,y
\end{aligned}
\tag{7}
$$

$$
\begin{aligned}
E_{z2} &= x*y*(U1 - U2 - Q) + x*(1-y)*(U1 + G2*j*n - U2) + (1-x)*y*(U1 - U2 - Q) \\
&\quad + (1-x)*(1-y)*(U1 - U2) \\
&= U1 - U2 - Qy + x(G2\,j\,n - G2\,j\,n\,y)
\end{aligned}
\tag{8}
$$

$$
\begin{aligned}
\bar{E}_z &= z*E_{z1} + (1-z)E_{z2} \\
&= U1 - Qy + G2\,j\,n\,x(-1+y)(-1+z) + P\,z + (-P + Q + G1\,j\,v\,x)y\,z - U2(1 \\
&\quad + g\,z)
\end{aligned}
\tag{9}
$$

According to the Malthusian dynamic equation, the rate of change of a strategy is equal to its degree of adaptation [50]. Therefore, the differential equations of the replication dynamic process of the GEUS evolutionary game can be expressed based on the results of the three-party expected return equation. The replication dynamic equations of the three parties are

shown below.

$$F_{(x)} = dx/d_t = x(E_{x1} - \bar{E}_x) = (-1 + x)x(G2j - G1\,k\,y\,z) \tag{10}$$

$$\begin{aligned} F_{(y)} = dy/d_t &= y(E_{y1} - \bar{E}_y) \\ &= (-1 + y)y(E2\,f + G2\,j\,m\,x + Q(-1 + z) - (P + G2\,j\,m\,x + G1\,k\,u\,x)z) \end{aligned} \tag{11}$$

$$\begin{aligned} F_{(z)} = dz/d_t &= z(E_{z1} - \bar{E}_z) \\ &= (g\,U2 + G2\,j\,n\,x + P(-1 + y) - (Q + G2\,j\,n\,x + G1\,j\,v\,x)y)(-1 + z)z \end{aligned} \tag{12}$$

The evolutionary stability strategy (ESS) of a system of differential equations can be discovered using Friedman's approach from the local stability analysis of the system's Jacobi matrix [51]. Below is a diagram of the triangular evolutionary Jacobi matrix.

$$J = \begin{pmatrix} \dfrac{\partial F_{(x)}}{\partial x} & \dfrac{\partial F_{(x)}}{\partial y} & \dfrac{\partial F_{(x)}}{\partial z} \\[2mm] \dfrac{\partial F_{(y)}}{\partial x} & \dfrac{\partial F_{(y)}}{\partial y} & \dfrac{\partial F_{(y)}}{\partial z} \\[2mm] \dfrac{\partial F_{(z)}}{\partial x} & \dfrac{\partial F_{(z)}}{\partial y} & \dfrac{\partial F_{(z)}}{\partial z} \end{pmatrix} = \begin{pmatrix} H_1 & H_2 & H_3 \\ H_4 & H_5 & H_6 \\ H_7 & H_8 & H_9 \end{pmatrix} \tag{13}$$

$$H_1 = (-1 + 2x)(G2j - G1kyz)$$

$$H_2 = -G1k(-1 + x)x\,z$$

$$H_3 = -G1k(-1 + x)x\,y$$

$$H_4 = (-1 + y)y(-G2\,j\,m(-1 + z) - G1\,k\,u\,z)$$

$$H_5 = (-1 + 2\,y)(C2\,t + G2\,j\,m\,x + Q(-1 + z) - (P + G2\,j\,m\,x + G1\,k\,u\,x)z)$$

$$H_6 = -(P - Q + G2\,j\,m\,x + G1\,k\,u\,x)(-1 + y)y$$

$$H_7 = -j(G2\,n(-1 + y) + G1\,v\,y)(-1 + z)z$$

$$H_8 = (P - Q - j(G2\,n + G1\,v)x)(-1 + z)z$$

$$H_9 = (g\,S2 + G2\,j\,n\,x + P(-1 + y) - (Q + G2\,j\,n\,x + G1\,j\,v\,x)$$

Since the system of replicated dynamic equations in evolutionary games represents the system's dynamics, the equilibrium point determined from the dynamic equations of individual participating subjects is not always the system's evolutionary stable strategy. If a specific strategy used by a game subject is a stable state, the probability x, y, and z of the three players who

**Table 3. Stability discriminant results of GEUS evolutionary strategy.**

| Strategy | Stability | Symbol | $\lambda_1$ | $\lambda_2$ | $\lambda_3$ |
|---|---|---|---|---|---|
| S1(1,1,1) | Yes | -, -, - | $G2\,j{-}G1\,k$ | $-P{+}E2\,f$ | $-Q{+}g\,U2{-}G1\,j\,v$ |
| | | | | $-G1\,k\,u$ | |
| S3(1,1,0) | No | +, *, * | $G2\,j$ | $G2\,j\,m{-}Q{+}E2\,f$ | $Q{-}g\,U2{+}G1\,j\,v$ |
| S5(1,0,1) | No | +, *, * | $G2\,j$ | $G2\,j\,n{-}P{+}g\,U2$ | $P{-}E2\,f{+}G1\,k\,u$ |
| S7(1,0,0) | No | +, *, * | $G2\,j$ | $-G2\,j\,n{+}P{-}g\,U2$ | $-G2\,j\,m{+}Q{-}E2\,f$ |
| S2(0,1,1) | No | +, *, * | $-G2\,j{+}G1\,k$ | $-Q{+}g\,U2$ | $-P{+}E2\,f$ |
| S4(0,1,0) | Not Sure | -, *, * | $-G2\,j$ | $Q{-}g\,U2$ | $-Q{+}E2\,f$ |
| S6(0,0,1) | Not Sure | -, *, * | $-G2\,j$ | $-P{+}g\,U2$ | $P{-}E2\,f$ |
| S8(0,0,0) | Not Sure | -, *, * | $-G2\,j$ | $P{-}g\,U2$ | $Q{-}E2\,f$ |

Note

* stands for uncertainty

choose that approach must satisfy the following conditions:

$$F_{(x)} = 0, \frac{\partial F_{(x)}}{\partial x} < 0$$

$$F_{(y)} = 0, \frac{\partial F_{(y)}}{\partial y} < 0 \tag{14}$$

$$F_{(z)} = 0, \frac{\partial F_{(z)}}{\partial z} < 0$$

Only stable strategies are of value for research [52]. Therefore, after I have obtained the evolutionary strategies of the three parties, it is necessary to analyze the stability of the approach. Consistent with the conclusions of Ritzberger and Weibull, for the stable strategy in the evolutionary three-party game, only the stability of the eight strategies listed in Table 2 needs to be examined [53]. The sign of the eigenvalues of the Jacobi matrix can be used to determine the stability of a strategy based on Liapunov's stability theory [54]. If the sign of the eigenvalue $\lambda$ is completely negative, the strategy is stable; if it is positive, the approach is unstable. Table 3 displays the results of this paper's stability identification, which show that there is one stable strategy S1, and three uncertain strategies S4, S6, and S8.

To satisfy Hypothesis 3, I assume that the value of the motivated participation benefit minus the non-participation benefit for each party to participate should be more than 0. Because the parties can be encouraged to participate in this manner, and Table 4 lists the evaluation standards for each party.

**Table 4. General judgment criteria of GEUS.**

| Participants | Benefit of participation | Benefit of non-participation | Judgment criteria |
|---|---|---|---|
| Government | $G1(1{+}k){-}G2(1{+}j)$ | $G1{-}G2$ | $-G2\,j{+}G1\,k{>}0$ |
| Enterprise | $E1{+}G1ku{-}E2(1{+}f)$ | $E1{-}E2{-}P$ | $P{-}E2\,f{+}G1\,k\,u{>}0$ |
| University | $U1{+}G1jv{-}U2(1{+}g)$ | $U1{-}U2{-}Q$ | $Q{-}g\,U2{+}G1\,j\,v{>}0$ |

**Table 5. Rational explanation of the evolutionary strategy.**

| Strategy | $\lambda$ | Explanation |
|---|---|---|
| S1 (1,1,1) | $G2j-G1k<0$ | For government, the additional advantage of government engagement outweighs the additional cost of government non-participation. |
| | $-P+E2f$ $-G1ku<0$ | For enterprises, the additional expenditure for enterprise participation is less than the sum of the profit distributed by the government and the compensation to be given to universities for enterprise non-participation. |
| | $-Q+gU2$ $-G1jv<0$ | For universities, the additional expenditure for university participation is less than the sum of the benefits allocated by the government and the compensation to be given to enterprises for university non-participation. |
| S4 (0,1,0) | $-G2j<0$ | Government merely spends; there is no profit. |
| | $Q-gU2<0$ | The compensation to be paid to the enterprise for the non-participation of the university is smaller than the additional expenses of the university. |
| | $-Q+E2f<0$ | The compensation to be paid to the enterprise when the university does not participate is greater than the additional expenses of the enterprise. |
| S6 (0,0,1) | $-G2j<0$ | Government merely spends; there is no profit. |
| | $-P+g\,U2<0$ | The compensation to the university when the enterprise does not participate is greater than the additional expenses when the university participates. |
| | $P-E2\,f<0$ | The compensation to the university when the enterprise does not participate is less than the additional expenses when the enterprise participates. |
| S8 (0,0,0) | $-G2j<0$ | Government merely spends; there is no profit. |
| | $P-g\,U2<0$ | The additional expenditure when the university participates is greater than the compensation to the university when the enterprise does not participate. |
| | $Q-E2\,f<0$ | The additional expenditure when the enterprise participates is greater than the compensation to the enterprise when the university does not participate. |

## 4. Simulation analysis

### 4.1 Theoretical analysis of evolutionary strategies

As indicated in Table 5, the stability points in Table 4 in this study are split into four distinct situations, S1, S4, S6, and S8. The government is only involved in S1. Because the government has only the increased expenditure $G2j$ in S4, S6, and S8, the government chooses not to engage in such three strategies.

In S1, the government prefers to participate only if the value of involvement outweighs the cost of participation. Similarly, for enterprises and universities to opt to participate, the additional costs would have to be lower than the advantages received. It is interesting to note that enterprises and universities not only consider the benefits they can share if GEUS succeeds, but also the costs they must pay if they default. This logic of dialectical thinking is in line with the reality of the situation.

In S4, $E2f<Q<gU2$, enterprises believe that their benefits are protected, and even if the university defaults, its default damages $Q$ can be able to cover the additional expenses $E2f$ resulting from the enterprise's participation. Therefore, enterprises will choose to participate. Nevertheless, universities decide not to participate to save money because the cost $Q$ of doing so is less than the additional expense $gU2$ that results from their participation.

In S6, $gU2<P<E2f$, the universities believe that their benefits are guaranteed, and even if the enterprise does not participate, its payout of default $P$ can cover the additional expenses incurred by the universities' participation. Thus, universities choose to participate. In contrast, the enterprise decides not to participate to cut costs because the cost $P$ incurred is less than the additional expense $E2f$ incurred when the enterprise participates.

In S8, $P<gU2$, the compensation given to the university in case of the enterprise's default is not enough to cover the additional expenses incurred by the university's participation, so the

university chooses not to participate. Similarly, $Q<E2f$, the enterprise decides not to participate since the compensation it could receive if the university defaults are insufficient to offset the higher costs associated with its involvement.

## 4.2 Simulation analysis of evolutionary strategies

With visible findings, simulation analysis can demonstrate the value that cannot be discovered through theoretical research [55]. The simulation data used in this study came from the GEUS study conducted in Beijing, China's capital. The data in this paper comes from the "Innovation and Entrepreneurship and University Industry-University-Research Report" of the China Business Intelligence Website, and the research data extracted from this report is shown in Table 6 [56]. Beijing has been at the forefront of China's GEUS cities since 2007 when Beijing issued the "Opinions on Encouraging Government-Enterprise-University Cooperation between Enterprises and Universities and Research Institutes in Beijing". In addition, Beijing's fiscal expenditure on GEUS has also increased year by year as a percentage of Beijing's regional GDP. It is reported that Beijing's GEUS expenditure has accounted for 6.44% of the regional GDP, while other cities are still less than 3%. Beijing also relies on the capital city's political, cultural, and information advantages, which unite several businesses and academic institutions to promote GEUS. The accomplishments of Beijing in GEUS are further highlighted by the success of ZOL Science Park, Qidi Science Park, and Peking University Science Park. Therefore, conducting a study using GEUS data from Beijing is both representative and illuminating.

Since S1 is the determined stabilization strategy, the evolutionary parameters involved in the replication dynamic equations are set in this paper based on the eigenvalue determination criteria of S1 and the current status of GEUS in Beijing, as shown in Table 6. It is worth emphasizing that all data units are in billion yuan except for the proportional data.

The simulation values are brought into Matlab software, the initial values of $x$, $y$, and $z$ are changed from 0.5 to 0.1, the termination value is set to 1, and the step size is set to 0.1. The evolution results obtained under the condition that the simulation time is 30 units are shown in Fig 4. It can be observed that the evolutionary trajectory of the tripartite participation in GEUS

**Table 6. Simulation parameters.**

| Participants | Parameter | Value |
|---|---|---|
| Government | G1 | 20 |
| | G2 | 10 |
| | k | 0.1 |
| | j | 0.1 |
| | u | 0.2 |
| | v | 0.8 |
| | m | 0.5 |
| | n | 0.5 |
| | x | 0.5 |
| Enterprise | E2 | 12 |
| | P | 6 |
| | f | 0.5 |
| | y | 0.5 |
| University | U2 | 8 |
| | Q | 6 |
| | g | 0.3 |
| | z | 0.5 |

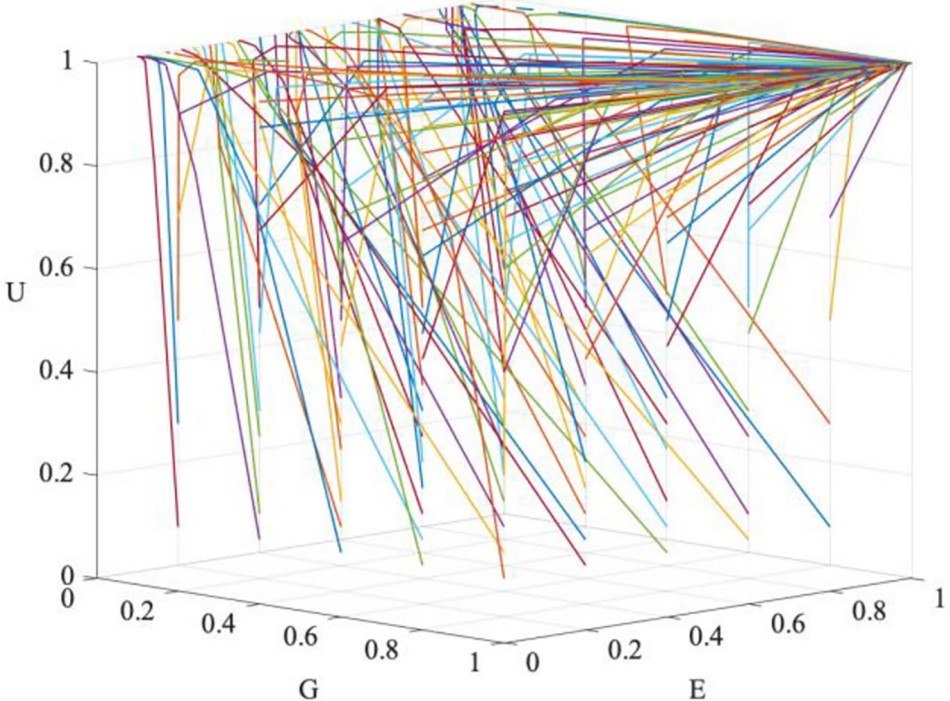

**Fig 4. Simulated evolutionary trajectory of GEUS.**

transitions from the initial S8 to S6, then shifts from S6 to S4, and finally converges on S1. On the one hand, the evolutionary trajectory demonstrates the validity of the simulation parameters set in this paper. On the other hand, it shows that the three parties in the GEUS make decisions in a dynamic manner that are consistent with the four scenarios shown in Table 5.

Initially, no one was willing to participate, while universities were the first to show a willingness to participate, and then enterprises began to have the motivation to participate, eventually, the government joined GEUS in response to the wishes of universities and enterprises to actively regulate. The combined efforts of the three parties will increase the size of the GEUS "cake," allowing all three to increase revenue while fostering innovation and addressing employment issues.

## 5. Results and discussion

### 5.1 Effect of initial willingness on evolutionary outcomes

The parameters are introduced into Matlab for simulation in this research to examine the effect of the three parties' initial willingness to engage in the evolutionary results. The initial values for $x$, $y$, and $z$ are 0.5, the termination values are 1, the step size is 0.1, and the simulation duration is 20 units. A willingness to participate of 0.5 indicates average participation, above 0.5 indicates positive participation, and below 0.5 indicates negative participation. The effect of willingness to participate on the evolutionary outcome is shown in Fig 5.

When the initial willingness of the three parties to participate is moderate, the system converges to point S1. Universities are the fastest to make participation decisions, followed by enterprises and finally the government. In the early stage of the evolution, the government's willingness to participate was relatively low, and even negative participation was observed. But in the middle stage, the government's willingness to participate began to exceed that of

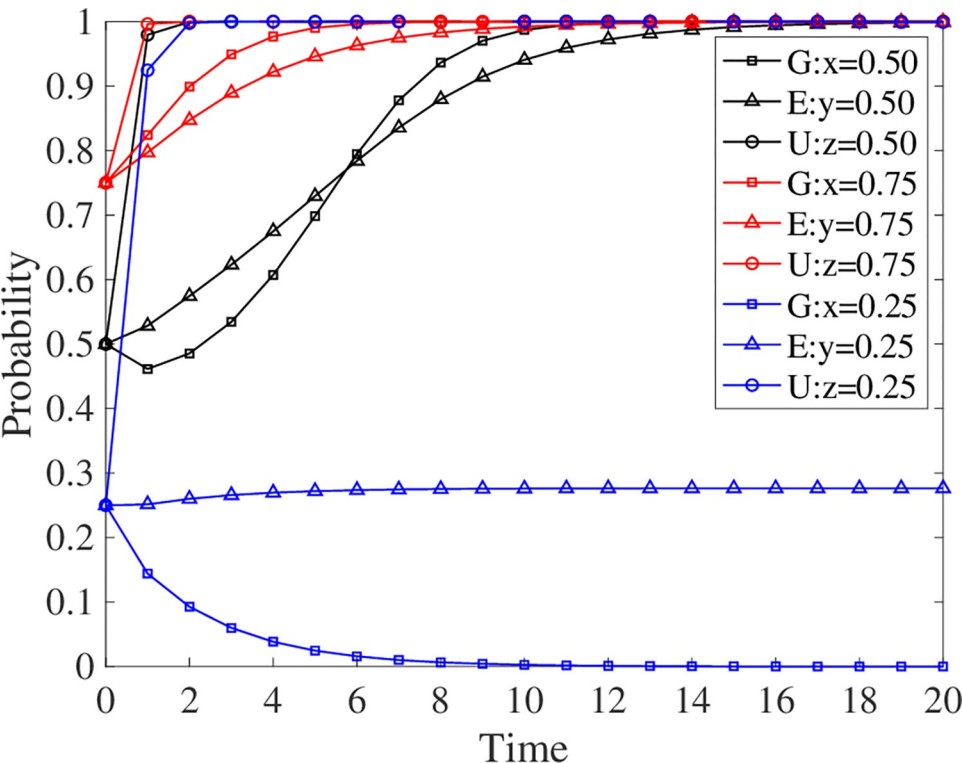

**Fig 5. Evolutionary strategy when the willingness of three parties to participate changes.**

enterprises, and eventually, all three parties chose to participate. In addition, when all three parties actively participate, the system converges into a stable strategy S1 more quickly. The intensity of willingness to participate is in the order of universities, government, and enterprises, from highest to lowest. Finally, when all three participants participate negatively, the system converges to S6. Universities continue to actively participate, while enterprises and the government begin to participate adversely. Overall, universities are the most eager to engage, owing to their desire to improve the quality and skills of talent training through GEUS. Enterprises and the government, on the other hand, are prone to non-participation. Since enterprises strive to maximize profits and rarely make uncertain judgments. When one party participates negatively, the government, as a regulator, is less eager to participate.

Although Fig 5 indicates that when the three parties are actively involved, the system converges to a stable strategy of S1 faster, the interaction of the three parties' willingness to participate remains a mystery. Therefore, the interaction process of the three parties in negative participation is handled independently in this research, as illustrated in Fig 6.

Figure (A) modifies the government's initial readiness to cooperate. It can be seen that as the government's initial desire wanes, the evolutionary decision of the three parties shifts from S1 to S6. The weakening of the government's willingness to participate does not affect universities, but it has a remarkable impact on enterprises' willingness to participate. When the government's willingness to participate falls from 0.5 to 0.25, both the enterprise's and the government's choice to participate is delayed, and the evolutionary intersection between the government and the enterprise rises from 0.8 to 0.85. This suggests that the government makes a deliberate decision to participate only when the enterprise demonstrates a greater readiness to cooperate. When the government's initial desire to participate falls below 0.25, it does not participate at all, while enterprises maintain a constant wait-and-see attitude.

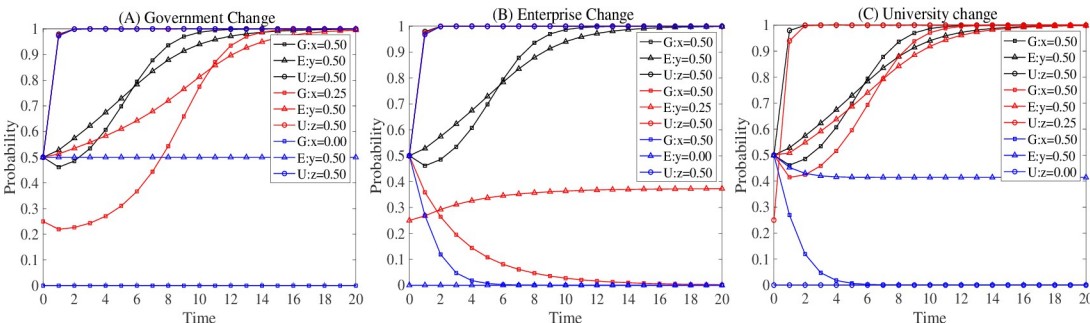

**Fig 6. Evolutionary strategy when a party's willingness to participate changes.**

Figure (B) adjusts the enterprise's initial readiness to engage. When the enterprise's initial willingness to cooperate falls, the evolutionary decision of the three parties goes from S1 to S6. While universities' readiness to participate is unaffected, the government's desire to cooperate varies widely. When the enterprise's initial desire to participate drops from 0.5 to 0.25, the government's willingness to participate shifts from participation to non-participation. Although an enterprise's intentions to participate negatively will shrink over time, it will never exhibit a willingness to actively participate. When enterprises' initial desire to participate declines from 0.25 to 0, the government decides not to participate faster. This implies that the greater the negative participation of enterprises, the sooner the government decides not to participate.

Figure (C) alters the universities' initial intention to participate. I discovered that as enterprises' initial desire to engage declines, the participation decision of the three parties shifts from S1 to S4. Enterprises are unaffected by universities' lessened inclination to participate, but governments are. When universities' initial intention to participate decreases from 0.5 to 0.25, it takes longer for the government and enterprises to make participation decisions. However, the willingness to participate in the evolutionary intersection of government and enterprises is unchanged, remaining near 0.8. When universities' initial willingness to participate reduces from 0.25 to 0, the government chooses non-participation, and enterprises begin to display a mildly negative willingness to participate and maintain it constantly.

## 5.2 Effect of default on evolutionary outcomes

To examine the effect of the default amount on the evolutionary results, this paper presents the evolutionary results at a default amount of 4, 6, and 8 (see Fig 7). It can be seen that the system converges to the stable strategy S1 faster when the default value increases. In contrast, the system converges to strategy S6 when the default value is decreased. It turns out that the level of default affects the evolutionary strategy. The higher the default, the more willing all three parties are to participate, with universities, corporations, and the government leading in descending order. And the government no longer seems to hesitate but makes a final decision to participate. Conversely, it takes longer for universities to decide to participate when the default is lowered, while businesses and the government pull back. These observations suggest that an increase in defaults drives GEUS, while a decrease in defaults limits GEUS.

## 5.3 Effect of benefit distribution ratio on evolutionary outcomes

As a rule, the government will allocate a larger share of the revenue to universities since they do not have the necessary social revenue [57]. In addition, enterprises receive a smaller portion of the revenue in recognition of their active participation [58]. However, it is unclear whether

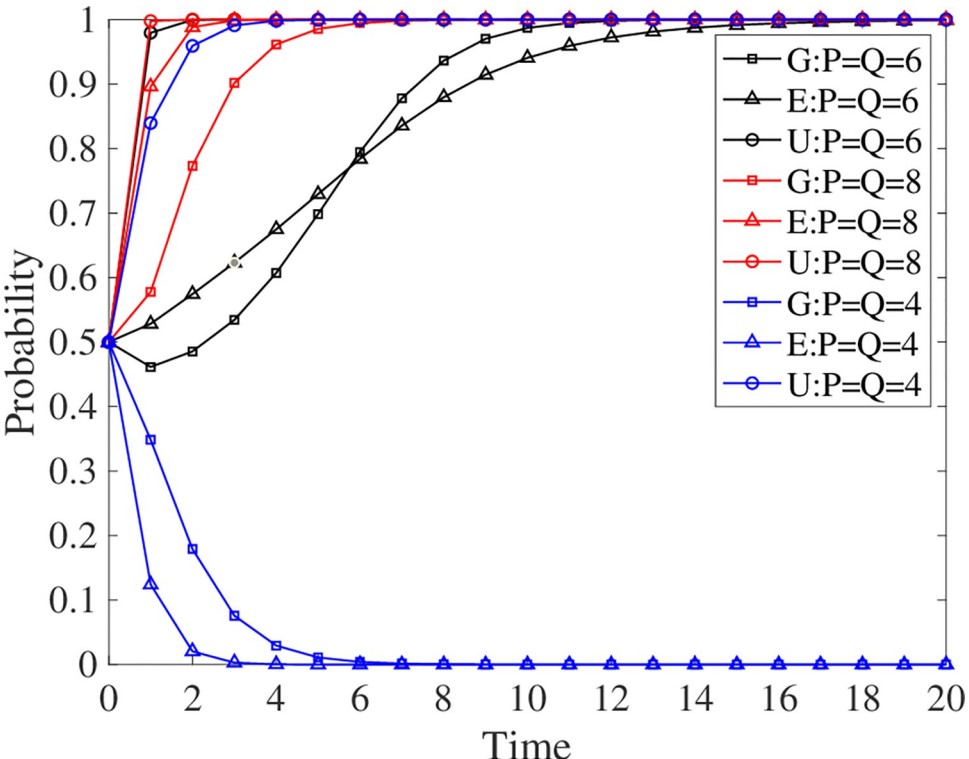

**Fig 7. Evolutionary strategy when the default amount changes.**

this setting is appropriate. To determine how the distribution of benefits affects development, this paper changes the proportion of benefits that enterprises and universities can receive. The results are shown in Fig 8. It seems that the share of government benefits does not affect the participation of universities in decision-making, but it affects the speed of participation of enterprises and governments. The higher the enterprise's share of the benefit distribution, the less time it takes the three parties to reach a stable strategy. Thus, if the government seeks to accelerate GEUS, it should tend to allocate more benefits to enterprises.

## 5.4 Effect of incentive allocation ratio on evolutionary outcomes

When incentivizing universities and enterprises to participate, the government typically uses equal incentive ratios to ensure fairness [59]. It is unknown, however, whether a skewed incentive ratio will hasten system evolution. Hence, this paper investigates the effect of incentive ratio changes on evolution, with the results shown in Fig 9. Changes in incentive ratios have a minor effect on evolution, and decreasing the incentive ratio distributed to enterprises can slightly accelerate evolution. Nevertheless, as the proportion of incentives distributed to enterprises grows, the system's evolutionary process slows. This appears to imply that adjustments in incentive ratios do not affect university engagement strategies but only slightly alter enterprise and government engagement decisions.

## 5.5 Effects of income and expenditure on evolutionary outcomes

Fig 10 depicts the impact of government revenues and expenditures on development outcomes. The left panel shows the impact of government revenues, while the right panel shows the impact of government expenditures. A comparison of the two graphs shows that both an

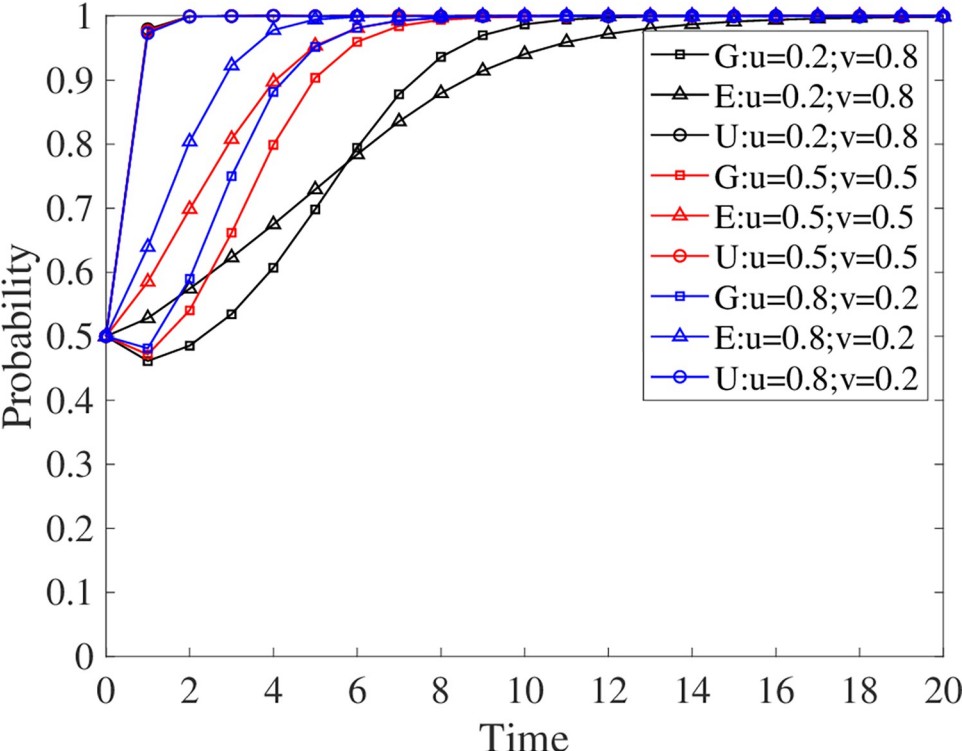

**Fig 8. Evolutionary strategy when the allocation ratio changes.**

increase in government revenues and a decrease in expenditures can accelerate the overall development of the system. The lower the government revenues are, the longer the three parties hesitate with their participation decisions. In contrast, excessive government spending leads to a shift in the stabilization strategy of the three parties. Excessive spending places a fiscal burden on the government that can cause a reversal in decision-making.

Fig 11 displays how enterprise and university expenditures affected the evolutionary outcomes. On the left is the impact of enterprise expenditure, and on the right is the influence of university expenditure. Contrast the two pictures, reducing enterprise expenditure can hasten the convergence of the entire system, whereas reducing university expenditure has little effect on the evolution. Therefore, it is the reduction in enterprise expenditure rather than the reduction in university expenditure that will push the system to converge faster.

## 6. Conclusion

### 6.1 Contributions of the study

This paper's scientific worth stems from theoretical supplementation on the one hand and practical instruction on the other. Theoretically, this dissertation provides a game model of educational collaboration that incorporates the government, enterprises, and universities, which compensates for the flaws of overlooking the government as a synergistic function in educational cooperation research. In practice, this paper offers specific counseling methods for encouraging the three parties to participate in educational cooperation, as well as establishing a mutually advantageous and win-win situation in educational cooperation.

**6.1.1 Theoretical contributions.** The theoretical contribution of this work lies in the new findings and the construction of a new research model. On the one hand, it highlights the

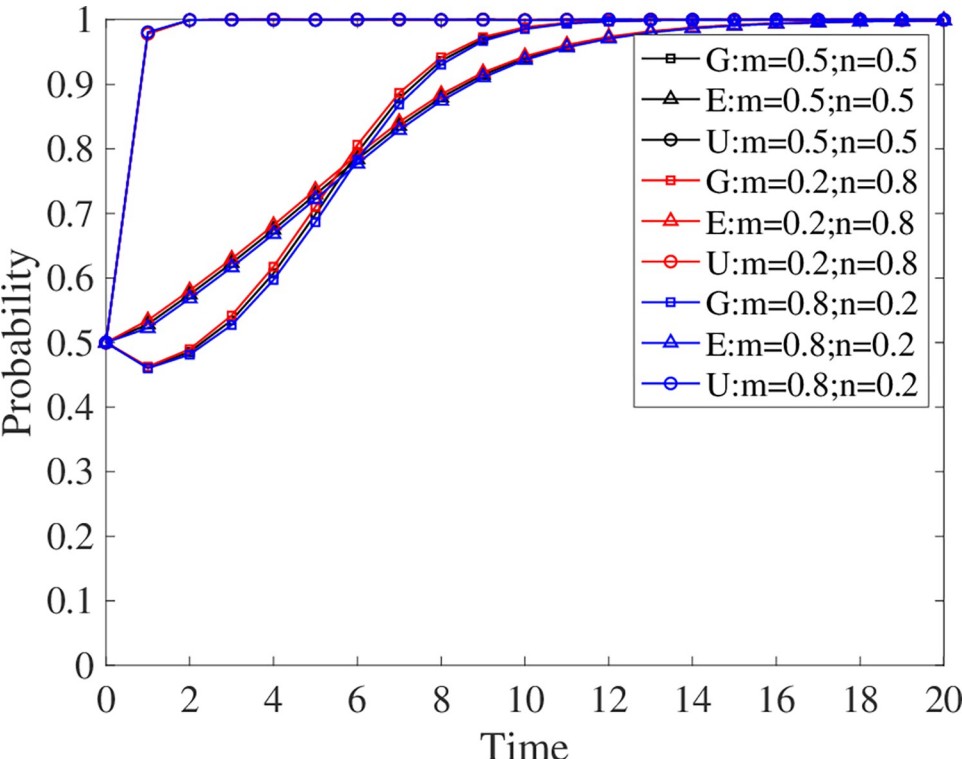

**Fig 9. Evolutionary strategy when the incentive ratio changes.**

differences in the willingness of government, business, and universities to participate in educational collaborations and corrects the traditional view that government is most likely to dominate educational collaborations. Previous studies usually assume that the government should have the strongest willingness to participate, as enterprises and universities only participate in GEUS under the leadership of the government [60]. However, this research comes to a different conclusion. In this paper, it is clear that the universities have the greatest desire to participate in GEUS, followed by the enterprises and then the government. The government is in a good position to act as a motivator and facilitator when both universities and enterprises want to participate.

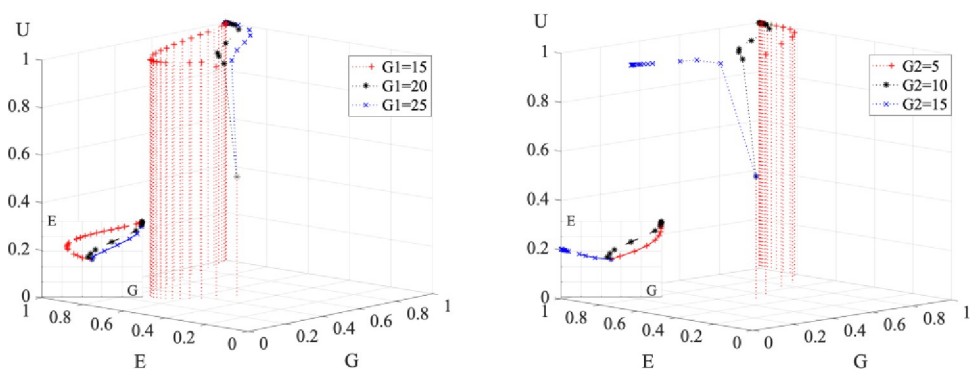

**Fig 10. Effect on evolutionary results when government revenues and expenditures change.**

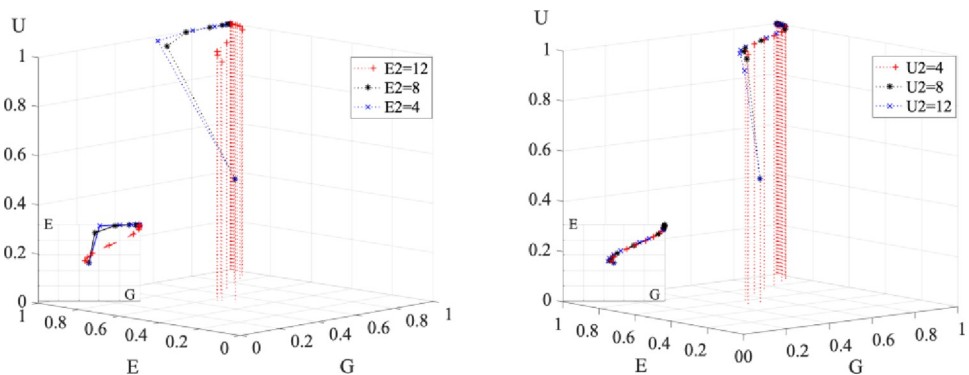

**Fig 11. Effect on evolutionary results when corporate and university expenditures change.**

On the other hand, this paper constructs a symbiotic evolutionary game model of government-enterprise-academia symbiosis that integrates the indicators of fusion revenue distribution ratio, which complements the indicator system of traditional research on government participation behavior. The current GEUS studies view the government as both an encourager and a punisher [16]. Therefore, only the government's function as an inducement and the punishment are taken into account. However, this article argues that the government will also play the role of the symbiont. If GEUS is successful, the government will allocate additional revenue to enterprises and universities, thus promoting the three together to make the "cake" of GEUS bigger. There is no longer a simple zero-sum game between the three parties, but win-win cooperation. Therefore, this study introduces the benefit allocation ratio. It is proved that GEUS can indeed be promoted by adjusting the benefits allocation ratio, and the larger the enterprise's share in the allocation ratio, the faster the three parties will choose GEUS.

**6.1.2 Practical contribution.** The practical contribution of this paper is to give specific strategies to promote better participation of government, enterprises, and universities in educational cooperation On the one hand, this paper gives strategies to accelerate educational cooperation in terms of the strength of the willingness of the three parties to participate in it, showing that active participation of the three parties can accelerate the evolution of the system, while passive participation can achieve the opposite effect. When the initial willingness of government and enterprises to participate changes, it does not affect the participation strategies of universities, but the effect between the two is particularly dramatic. Specifically, enterprises are the most inclined to a wait-and-see attitude, while the government is the most likely to opt not to participate. Furthermore, if the government's initial readiness to cooperate wanes, the government will first analyze enterprises' willingness to join. Only if enterprises are more willing to engage than before will the government opt to participate. Finally, the deeper the desire of enterprises to participate adversely, the greater the likelihood that the government will opt not to participate.

On the other hand, this paper provides three specific ways to accelerate educational cooperation in terms of incentives, penalties, and distribution mechanisms, which provide a grounded program for participants to better engage in educational cooperation. First, GEUS is unaffected by incentive allocation ratios. The government should focus on how to enhance the enterprise's benefit distribution ratio rather than focusing on how to set the incentive ratio. Because enterprises seek to maximize their economic benefits, changes in earnings distribution will trigger quick changes in their desire to participate. By contrast, universities are more concerned with academic values and social interests, along with talent development, scientific research, and social services [15]. Second, higher default fees should be imposed to expedite

the GEUS process, too low default fees will eliminate the incentive for enterprises and governments to participate. The government can employ negative incentives to lead enterprises and universities to participate in GEUS by appropriately increasing the penalty for default. Third, increasing government revenues while decreasing government and enterprise expenditure can hasten GEUS, whereas cutting university expenditure has little influence on GEUS. Therefore, the government should establish a municipal GEUS fund to share risk with businesses, decreasing costs on both sides. Implement tax advantages for enterprises who are willing to actively engage, cutting their costs.

## 6.2 Research limitations and future work

The study is bound to have limitations because it is an exploratory paper. Although the GEUS symbiotic evolutionary model developed in this study attempted to motivate the three participants to make cooperative and synergistic decisions, it is evident that graduates play a vital role in the process [61, 62]. This article makes no assumptions about the choices that graduates will make when confronted with university training environments and enterprise employment options. Further studies, which take graduates into account, will need to be undertaken. Despite its limitations, the study certainly adds to our understanding of the GEUS in cooperative education.

## Author Contributions

**Conceptualization:** Shuangzhi Zhang.

**Data curation:** Shuangzhi Zhang.

**Formal analysis:** Shuangzhi Zhang.

**Funding acquisition:** Shuangzhi Zhang.

**Investigation:** Shuangzhi Zhang.

**Methodology:** Shuangzhi Zhang.

**Project administration:** Shuangzhi Zhang.

**Resources:** Shuangzhi Zhang.

**Software:** Shuangzhi Zhang.

**Supervision:** Shuangzhi Zhang.

**Validation:** Shuangzhi Zhang.

**Visualization:** Shuangzhi Zhang.

**Writing – original draft:** Shuangzhi Zhang.

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
