## [Decision Letter · Decision Letter 0]

21 Aug 2023

PONE-D-23-23142Educational cooperation practice based on three-way evolutionary game simulationPLOS ONE

Dear Dr. zhang,

Thank you for submitting your manuscript to PLOS ONE. After careful consideration, we feel that it has merit but does not fully meet PLOS ONE’s publication criteria as it currently stands. Therefore, we invite you to submit a revised version of the manuscript that addresses the points raised during the review process.

We look forward to receiving your revised manuscript.

Kind regards,

Tinggui Chen

Academic Editor

PLOS ONE

Journal Requirements:

   "This study is supported by “MOE (Ministry of Education in China) Project of Humanities and Social Sciences” (Project No.22YJC630207)" 

Additional Editor Comments:

Submitting your manuscript to PLOS ONE.

I have completed my evaluation of your manuscript. The reviewers recommend reconsideration of your manuscript following major revision. I invite you to resubmit your manuscript after addressing the comments below.

Reviewers' comments:

Reviewer's Responses to Questions

**Comments to the Author**

1. Is the manuscript technically sound, and do the data support the conclusions?

Reviewer #1: Partly

Reviewer #2: No

2. Has the statistical analysis been performed appropriately and rigorously? 

Reviewer #1: Yes

Reviewer #2: No

3. Have the authors made all data underlying the findings in their manuscript fully available?

Reviewer #1: No

Reviewer #2: No

4. Is the manuscript presented in an intelligible fashion and written in standard English?

Reviewer #1: Yes

Reviewer #2: Yes

5. Review Comments to the Author

Reviewer #1: The reviewer believes that the topic “Educational cooperation practice based on three-way evolutionary game simulation” is worthy of investigation. However, the following needs to be addressed. There are minor and major issues that should be corrected. I believe the paper could be further strengthened by added information about.

1.The title does not provide a core theme of the topic.

2.Please specify the source of the simulation data.

3.The language of this manuscript needs help from native speakers.

4.Please underscore the scientific value-added to your paper in your abstract. Your abstract should clearly state the essence of the problem you are addressing, what you did and what you found and recommend. That would help a prospective reader of the abstract to decide if they wish to read the entire article.

5.Although some academics have begun to integrate....... This a very vague statement. These sentences do not provide any information on how the concept could be conceptualized? - The Introduction should have 1) a concise but complete justification of the topic's importance both academically and practically, and 2) an explanation of the gaps both in research and practice. Please review appropriate literature in the Introduction, with the research question clearly arising from that review.

6.-There is no flow in the text. It partly depends on the lack of proofreading but also on the fact that many statements and claims are made without being followed up by a clear and logical discussion. It is especially problematic in the Introduction that brings up a number of findings from different areas without linking them together.

7.-More importantly, the choice of the variables should be explained in light of the theory and the prior literature on the topic. The arguments are simply relationships and causes very close to the replication of many studies dealing with the same thing. For example, Enhancing engineering ethics education (EEE) for green intelligent manufacturing: Implementation performance evaluation of core mechanism of green intelligence EEE.

8.-Methodology: Model.. I suggest authors here build your main heading on Research and data methodology. Clearly explain the model building process, and what previous studies have used similar models (model testing approach).

See the following: A three-player game model for promoting the diffusion of green technology in manufacturing enterprises from the perspective of supply and demand.https://doi.org/10.3390/math8091585

A stochastic differential game of low carbon technology sharing in collaborative innovation system of superior enterprises and inferior enterprises under uncertain environment, https://doi.org/10.1515/math-2018-0056

9.Please make sure your conclusions' section underscores the scientific value-added of your paper, and/or the applicability of your findings/results. Highlight the novelty of your study. In addition to summarizing the actions taken and results, please strengthen the explanation of their significance. It is recommended to use quantitative reasoning comparing with appropriate benchmarks, especially those stemming from previous work.

Reviewer #2: It is a very interesting topic. Below presented some detailed comments. Hopefully they can assist the authors to improve the quality of the manuscript.

1. In Introduction, it was not stated the necessity that the Government-Enterprise-University Synergy requires the use of evolutionary games for research.

2. The authors should increase the literature review that the application research of evolutionary game methods on government-schools, government-enterprises, and schools-enterprises.

3. The authors have merely listed out the studies without even creating a debate among them. Without that debate and contradictions, the research gap cannot be reflected. These should be classified and presented.

4. The author should compare the results of relevant research in Results and Discussion.

5. The research contribution of this manuscript is relatively weak, and I hope the author can supplement it.

6. The main conclusion of this manuscript appears in the third point of 6.1.1 Theoretical contribution, and the management implications appear in 6.1.2 Practical contribution section. Can these two parts form a one-to-one correspondence? Otherwise, the management implications are not deep-going.

7. This manuscript needs careful editing by someone with expertise in technical English editing.

6. PLOS authors have the option to publish the peer review history of their article (what does this mean?). If published, this will include your full peer review and any attached files.

Reviewer #1: No

Reviewer #2: No

---

## [Author Response · Author response to Decision Letter 0]

10 Oct 2023

Dear Editors and Reviewers:

Thank you for your decision and constructive comments on my manuscript. I appreciate the editor and reviewers very much for their positive and constructive comments and suggestions on our manuscript entitled "Educational cooperation practice based on three-way evolutionary game simulation".

I have studied the reviewers’ comments carefully and have made revisions, which are marked in yellow in the revised manuscript. I have tried my best to revise our manuscript according to the comments. Attached, please find the revised version, which I would like to submit for your kind consideration. Revision notes, point-to-point, are given in the following tables.

I would like to express my great appreciation to you and the reviewers for their comments on our paper. I look forward to hearing from you.

Thank you, and best regards.

Yours sincerely,

ShuangZhi Zhang 

Manuscript Number: PONE-D-23-23142

Title: Educational cooperation practice based on three-way evolutionary game simulation

Reviewers' number Reviewers' 

comments Our responses

Reviewer #1 

1. The title does not provide a core theme of the topic.

Reply:

Thank you very much for your suggestion, I have revisited the topic of the whole paper and formulated a new title with reference to your article (Enterprises from the Perspective of Supply and Demand A Three-Player Game Model for Promoting the Diffusion of Green Technology in Manufacturing), I hope that you will be satisfied with my revision.

Modify:

(The Title)

Educational Cooperation in the Perspective of Tripartite Evolutionary Game among Government, Enterprises and Universities

2. Please specify the source of the simulation data. 

Reply:

My data comes from China Business Intelligence Website, I have extracted the parameter values about government, universities and enterprises from the report of Innovation and Entrepreneurship and University-Industry-Research 2024-2029, I have put the link of this report in the text in the form of reference, please check it.

Modify:

(The first paragraph of 4.2 Simulation analysis of evolutionary strategies)

The data in this paper comes from the "Innovation and Entrepreneurship and University Industry-University-Research Report" of the China Business Intelligence Website, and the research data extracted from this report is shown in Table 6 [56].

56. China BRI. Innovation and Entrepreneurship and University Industry-University-Research Report. China Business Intelligence Website. 2023. doi:https://www.askci.com/reports/20210917/1533425910902916.shtml

3. The language of this manuscript needs help from native speakers. 

Reply:

Your suggestions help to improve the quality of this article, I have invited a native speaker to touch up the processing and grammar check of this article, and I hope you can be satisfied with the processing.

4. Please underscore the scientific value-added to your paper in your abstract. Your abstract should clearly state the essence of the problem you are addressing, what you did and what you found and recommend. That would help a prospective reader of the abstract to decide if they wish to read the entire article. 

Reply:

I admire your academic rigor and I have processed the abstract of this paper according to the four points you have pointed out.

Modify:

(The Abstract)

Government-enterprise-university synergy (GEUS) is an effective way to mobilize government, enterprises, and universities to collaborate on education, but these three parties involved in GEUS may, out of bounded rationality, choose to collaborate in ways that benefit themselves and harm others. To guide the three parties to better cooperation, this study creates an evolutionary game model among the three parties and evaluates the applicability and validity of the model by selecting the educational cooperation data in Beijing. It is shown that participation in education cooperation is the best course of action for all three parties. The intensity of willingness to participate in the GEUS is on the order of high to low for universities, enterprises, and the government. If the three parties wish to accomplish education collaboration sooner, they can increase default payments, boost government revenues, raise corporate participation in distribution, and reduce government and government spending. These results highlight the inherent regularities of GEUS and provide concrete implementation strategies to improve the efficiency of education cooperation.

5. Although some academics have begun to integrate....... This a very vague statement. These sentences do not provide any information on how the concept could be conceptualized? - The Introduction should have 1) a concise but complete justification of the topic's importance both academically and practically, and 2) an explanation of the gaps both in research and practice. Please review appropriate literature in the Introduction, with the research question clearly arising from that review. 

Reply:

Thank you very much for such a detailed review, I learned a lot of writing points from your review, and I have reorganized the introduction section based on your two suggestions.

Modify:

(The fourth and fifth paragraph of 1. Introduction)

Currently, most of the existing studies on educational cooperation discuss only the interest game between enterprises and universities, ignoring the leading role of the government [10–12]. The government is more often considered as an exogenous variable in educational cooperation rather than being included as a member of the game involved [13]. This disconnect ignores the co-constructive role of government in education cooperation and fails to recognize that cooperation between government, enterprises, and universities is a synergistic and symbiotic system of government, enterprises, and colleges and universities.

To address the shortcomings of previous studies, this paper aims to incorporate the symbiotic nature of government into the tripartite evolutionary game system, viewing government as a co-constructor of educational cooperation rather than merely a facilitator. By constructing a tripartite evolutionary game model of education cooperation that includes the government in the symbiotic system, this paper not only identifies the optimal game strategies of the government, enterprises, and universities in education cooperation, but also provides concrete implementation suggestions to promote the participation of the three parties in education cooperation. Overall, the conclusions of this paper make up for the lack of game strategy research on the role of the government in educational cooperation in academia and provide a concrete program to strengthen the synergy between the government, enterprises and universities in practice.

11. Johnston A, Huggins R. Partner selection and university-industry linkages: Assessing small firms’ initial perceptions of the credibility of their partners. Technovation. 2018;78: 15–26. doi:10.1016/j.technovation.2018.02.005

12. Rajalo S, Vadi M. University-industry innovation collaboration: Reconceptualization. Technovation. 2017;62: 42–54. doi:10.1016/j.technovation.2017.04.003

13. Sarpong D, AbdRazak A, Alexander E, Meissner D. Organizing practices of university, industry and government that facilitate (or impede) the transition to a hybrid triple helix model of innovation. Technological Forecasting and Social Change. 2017;123: 142–152. doi:10.1016/j.techfore.2015.11.032

6. There is no flow in the text. It partly depends on the lack of proofreading but also on the fact that many statements and claims are made without being followed up by a clear and logical discussion. It is especially problematic in the Introduction that brings up a number of findings from different areas without linking them together.

Reply:

I have taken your comments very seriously, and I have revisited the inherent logical connections of the various findings in the introductory section and sorted out those connections, I hope you will be satisfied with my revisions.

Modify:

(The third paragraph of 1. Introduction) 

However, because the government, enterprises, and universities don't know what kind of decision each other will make when deciding whether or not to engage in educational cooperation, it's difficult to synchronize decision-making among the three, making it hard to promote educational cooperation smoothly [8]. The conduct of numerous parties involved in decision-making is perfectly rational in the classical game, and everyone makes well-considered decisions with completely open information [9]. However, because of the three cross-organizational levels and barriers, the condition of complete information does not apply to GEUS. Therefore, cooperation among government, enterprises, and universities is a constrained rational decision made by three parties under information asymmetry, in which the three parties gradually learn from each other's decisions and improve their own strategies based on each other's decisions. The three parties eventually developed a stable cooperative intention to maximize their own interests through continual modification in the conflict resolution process. As a result, the evolutionary game method should be used to investigate government-business-academic collaboration.

8. Wang C, Medaglia R, Zheng L. Towards a typology of adaptive governance in the digital government context: The role of decision-making and accountability. Government Information Quarterly. 2018;35: 306–322. doi:10.1016/j.giq.2017.08.003

9. Simon GA. Rational decision making in business organizations - Nobel memorial lection delivered in December 8, 1977. PSIKHOLOGICHESKII ZHURNAL. 2002;23: 42–51. 

7. More importantly, the choice of the variables should be explained in light of the theory and the prior literature on the topic. The arguments are simply relationships and causes very close to the replication of many studies dealing with the same thing. For example, Enhancing engineering ethics education (EEE) for green intelligent manufacturing: Implement. 

Reply:

Your suggestions help to highlight the value of this paper, and I have explained the rationale for the choice of variables in this paper and compared the differences between these variables and those chosen in previous studies. In addition, I have carefully read all of the articles you recommended and have incorporated them into this thesis for careful comparison.

Modify:

(The last paragraph of 3.1 The Parameter System of the GEUS Model)

It is worth emphasizing that this paper is inspired by Yin Shi's research on engineering ethics education [47]. In scholar Yin Shi's study on collaborative education, the author emphasizes that cooperative education is a combination of participating objects under certain rules and that this whole achieves educational synergy through resource sharing and optimal allocation. This viewpoint coincides with this paper's promotion of government, enterprises, and universities as a symbiotic system that promotes educational cooperation. Nevertheless, there are some differences between the evolutionary game indicator system constructed in this paper and the implementation effect evaluation system constructed by Yin Shi, who constructed an implementation effect evaluation system with 27 indicators from the perspective of cultivating education, collaborative education, and situational education, while this paper constructs an evolutionary game indicator system with 19 indicators from the perspective of participating members. In other words, Yin Shi's study discusses collaborative education through the subject's participation in scenarios, while this paper studies educational cooperation through the subject's participation in decision-making. In terms of research methodology, this paper is also inspired by Yin Shi's work on three-way games [48]. This author discusses the game among government, demand firms, and supply firms in technological innovation in the manufacturing industry, which inspires me to explore the game of educational cooperation among government, firms, and universities. In addition, Yin Shi's concept of symbiosis in collaborative innovation systems also provides support for this paper's attempt to construct a symbiosis system between government, enterprises, and universities [49]. However, the objective of this work is educational cooperation in education-oriented employment, which is distinct from the concept of collaborative invention that Yin Shi focuses on in game research.

47. Yin S, Zhang N. Enhancing engineering ethics education (EEE) for green intelligent manufacturing: Implementation performance evaluation of core mechanism of green intelligence EEE. FRONTIERS IN PSYCHOLOGY. 2022;13: 926133. doi:10.3389/fpsyg.2022.926133

48. Wang M, Lian S, Yin S, Dong H. A Three-Player Game Model for Promoting the Diffusion of Green Technology in Manufacturing Enterprises from the Perspective of Supply and Demand. MATHEMATICS. 2020;8: 1585. doi:10.3390/math8091585

49. Yin S, Li B. A stochastic differential game of low carbon technology sharing in collaborative innovation system of superior enterprises and inferior enterprises under uncertain environment. OPEN MATHEMATICS. 2018;16: 607–622. doi:10.1515/math-2018-0056

8. Methodology: Model. I suggest authors here build your main heading on Research and data methodology. Clearly explain the model building process, and what previous studies have used similar models (model testing approach). 

Reply:

Thanks to your selfless contribution, I have borrowed the title-setting methodology of the two articles you provided and changed the main title in this paper 3. The GEUS Evolutionary Game Model. In addition, I have cited the three articles you suggested in the last paragraph of 3.1 The Parameter System of the GEUS Model to thank you for your offer, and have highly valued their contributions to this paper, as well as explain the discrepancies that exist between these articles and this study.

Modify:

(The last paragraph of 3.1 The Parameter System of the GEUS Model)

It is worth emphasizing that this paper is inspired by Yin Shi's research on engineering ethics education [47]. In scholar Yin Shi's study on collaborative education, the author emphasizes that cooperative education is a combination of participating objects under certain rules and that this whole achieves educational synergy through resource sharing and optimal allocation. This viewpoint coincides with this paper's promotion of government, enterprises, and universities as a symbiotic system that promotes educational cooperation. Nevertheless, there are some differences between the evolutionary game indicator system constructed in this paper and the implementation effect evaluation system constructed by Yin Shi, who constructed an implementation effect evaluation system with 27 indicators from the perspective of cultivating education, collaborative education, and situational education, while this paper constructs an evolutionary game indicator system with 19 indicators from the perspective of participating members. In other words, Yin Shi's study discusses collaborative education through the subject's participation in scenarios, while this paper studies educational cooperation through the subject's participation in decision-making. In terms of research methodology, this paper is also inspired by Yin Shi's work on three-way games [48]. This author discusses the game among government, demand firms, and supply firms in technological innovation in the manufacturing industry, which inspires me to explore the game of educational cooperation among government, firms, and universities. In addition, Yin Shi's concept of symbiosis in collaborative innovation systems also provides support for this paper's attempt to construct a symbiosis system between government, enterprises, and universities [49]. However, the objective of this work is educational cooperation in education-oriented employment, which is distinct from the concept of collaborative invention that Yin Shi focuses on in game research.

47. Yin S, Zhang N. Enhancing engineering ethics education (EEE) for green intelligent manufacturing: Implementation performance evaluation of core mechanism of green intelligence EEE. FRONTIERS IN PSYCHOLOGY. 2022;13: 926133. doi:10.3389/fpsyg.2022.926133

48. Wang M, Lian S, Yin S, Dong H. A Three-Player Game Model for Promoting the Diffusion of Green Technology in Manufacturing Enterprises from the Perspective of Supply and Demand. MATHEMATICS. 2020;8: 1585. doi:10.3390/math8091585

49. Yin S, Li B. A stochastic differential game of low carbon technology sharing in collaborative innovation system of superior enterprises and inferior enterprises under uncertain environment. OPEN MATHEMATICS. 2018;16: 607–622. doi:10.1515/math-2018-0056

9. Please make sure your conclusions' section underscores the scientific value-added of your paper, and/or the applicability of your findings/results. Highlight the novelty of your study. In addition to summarizing the actions taken and results, please strengthen the explanation of their significance. It is recommended to use quantitative reasoning comparing with appropriate benchmarks, especially those stemming from previous work.

Reply:

Thank you very much for your attention to this paper, I have added the scientific value and practical significance of this paper in the conclusion section and emphasized the novelty of this paper in educational employment research, I hope you will be satisfied with my response.

Modify:

(The first paragraph of 6.1 Contributions of the study)

This paper's scientific worth stems from theoretical supplementation on the one hand and practical instruction on the other. Theoretically, this dissertation provides a game model of educational collaboration that incorporates the government, enterprises, and universities, which compensates for the flaws of overlooking the government as a synergistic function in educational cooperation research. In practice, this paper offers specific counseling methods for encouraging the three parties to participate in educational cooperation, as well as establishing a mutually advantageous and win-win situation in educational cooperation.

Reviewer #2 

1. In Introduction, it was not stated the necessity that the Government-Enterprise-University Synergy requires the use of evolutionary games for research. 

Reply:

Thank you for such valuable suggestions, I have added to the choices made by the parties in government, enterprises and universities under limited rationality, and it is this variability in decision-making that leads to differences in the outcomes of education and employment. To ensure that all three parties make decisions that are favorable to education and employment one must introduce game theory, which can reconcile cooperation and conflict between multiple players and allow the strategies of the players to evolve in the direction of favoring education and employment through trial and error.

Modify:

(The third paragraph of 1. Introduction) 

However, because the government, enterprises, and universities don't know what kind of decision each other will make when deciding whether or not to engage in educational cooperation, it's difficult to synchronize decision-making among the three, making it hard to promote educational cooperation smoothly [8]. The conduct of numerous parties involved in decision-making is perfectly rational in the classical game, and everyone makes well-considered decisions with completely open information [9]. However, because of the three cross-organizational levels and barriers, the condition of complete information does not apply to GEUS. Therefore, cooperation among government, enterprises, and universities is a constrained rational decision made by three parties under information asymmetry, in which the three parties gradually learn from each other's decisions and improve their own strategies based on each other's decisions. The three parties eventually developed a stable cooperative intention to maximize their own interests through continual modification in the conflict resolution process. As a result, the evolutionary game method should be used to investigate government-business-academic collaboration.

8. Wang C, Medaglia R, Zheng L. Towards a typology of adaptive governance in the digital government context: The role of decision-making and accountability. Government Information Quarterly. 2018;35: 306–322. doi:10.1016/j.giq.2017.08.003

9. Simon GA. Rational decision making in business organizations - Nobel memorial lection delivered in December 8, 1977. PSIKHOLOGICHESKII ZHURNAL. 2002;23: 42–51.

2. The authors should increase the literature review that the application research of evolutionary game methods on government-schools, government-enterprises, and schools-enterprises. 

Reply:

I appreciate your suggestion, but I am not quite sure what part of the text you would like us to add what you call applied research to, whether it is a literature review or research conclusions. Because my current version has already reviewed the application of game theory in educational cooperation in the third paragraph of 2.2 Evolutionary Gaming. In addition, what this paper addresses is the game of educational cooperation under a three-way synergy between government, enterprises, and universities, which I will consider here as a whole. If I follow your opinion to study the game strategy of each pair of game objects, I think it is not quite in line with the theme of this paper - tripartite evolutionary game. 

3. The authors have merely listed out the studies without even creating a debate among them. Without that debate and contradictions, the research gap cannot be reflected. These should be classified and presented. 

Reply:

I greatly admire your expertise and scholarship, I have summarized and categorized the previous studies and compared the differences between these studies and this paper, I hope you will be satisfied with my revisions.

Modify:

(The fourth and fifth paragraph of 1. Introduction)

Currently, most of the existing studies on educational cooperation discuss only the interest game between enterprises and universities, ignoring the leading role of the government [10–12]. The government is more often considered as an exogenous variable in educational cooperation rather than being included as a member of the game involved [13]. This disconnect ignores the co-constructive role of government in education cooperation and fails to recognize that cooperation between government, enterprises, and universities is a synergistic and symbiotic system of government, enterprises, and colleges and universities.

To address the shortcomings of previous studies, this paper aims to incorporate the symbiotic nature of government into the tripartite evolutionary game system, viewing government as a co-constructor of educational cooperation rather than merely a facilitator. By constructing a tripartite evolutionary game model of education cooperation that includes the government in the symbiotic system, this paper not only identifies the optimal game strategies of the government, enterprises, and universities in education cooperation, but also provides concrete implementation suggestions to promote the participation of the three parties in education cooperation. Overall, the conclusions of this paper make up for the lack of game strategy research on the role of the government in educational cooperation in academia and provide a concrete program to strengthen the synergy between the government, enterprises and universities in practice.

11. Johnston A, Huggins R. Partner selection and university-industry linkages: Assessing small firms’ initial perceptions of the credibility of their partners. Technovation. 2018;78: 15–26. doi:10.1016/j.technovation.2018.02.005

12. Rajalo S, Vadi M. University-industry innovation collaboration: Reconceptualization. Technovation. 2017;62: 42–54. doi:10.1016/j.technovation.2017.04.003

13. Sarpong D, AbdRazak A, Alexander E, Meissner D. Organizing practices of university, industry and government that facilitate (or impede) the transition to a hybrid triple helix model of innovation. Technological Forecasting and Social Change. 2017;123: 142–152. doi:10.1016/j.techfore.2015.11.032

4. The author should compare the results of relevant research in Results and Discussion. Reply:

Your idea is extremely helpful; the contribution and novelty of this study can only be highlighted by comparing it to past research results; I have included a comparison between this paper and earlier research in the text; I hope you are satisfied with the response.

Modify:

5.3 Effect of benefit distribution ratio on evolutionary outcomes

As a rule, the government will allocate a larger share of the revenue to universities since they do not have the necessary social revenue [57]. In addition, enterprises receive a smaller portion of the revenue in recognition of their active participation [58] However, it is unclear whether this setting is appropriate. To determine how the distribution of benefits affects development, this paper changes the proportion of benefits that enterprises and universities can receive. The results are shown in Figure 8. It seems that the share of government benefits does not affect the participation of universities in decision-making, but it affects the speed of participation of enterprises and governments. The higher the enterprise's share of the benefit distribution, the less time it takes the three parties to reach a stable strategy. Thus, if the government seeks to accelerate GEUS, it should tend to allocate more benefits to enterprises.

5.4 Effect of incentive allocation ratio on evolutionary outcomes

When incentivizing universities and enterprises to participate, the government typically uses equal incentive ratios to ensure fairness [59]. It is unknown, however, whether a skewed incentive ratio will hasten system evolution. Hence, this paper investigates the effect of incentive ratio changes on evolution, with the results shown in Fig. 9. Changes in incentive ratios have a minor effect on evolution, and decreasing the incentive ratio distributed to enterprises can slightly accelerate evolution. Nevertheless, as the proportion of incentives distributed to enterprises grows, the system's evolutionary process slows. This appears to imply that adjustments in incentive ratios do not affect university engagement strategies but only slightly alter enterprise and government engagement decisions.

57. Badelt C. Private external funding of universities: Blind alley or new opening? Review of Managerial Science. 2020;14: 447–458. doi:10.1007/s11846-019-00365-0

58. Milesi D, Verre V, Petelski N. Science-industry R&D cooperation effects on firm’s appropriation strategy: the case of Argentine biopharma. European Journal of Innovation Management. 2017. doi:10.1108/EJIM-07-2015-0058

59. Liu W, Yang J. The evolutionary game theoretic analysis for sustainable cooperation relationship of collaborative innovation network in strategic emerging industries. Sustainability. 2018;10: 4585. doi:10.3390/su10124585

5. The research contribution of this manuscript is relatively weak, and I hope the author can supplement it. 

Reply:

Your suggestions are crucial to improving the quality of this paper. I have reorganized the theoretical and practical contributions of this study, and I hope you will be satisfied with my revisions.

Modify:

(The first paragraph of 6.1 Contributions of the study)

This paper's scientific worth stems from theoretical supplementation on the one hand and practical instruction on the other. Theoretically, this dissertation provides a game model of educational collaboration that incorporates the government, enterprises, and universities, which compensates for the flaws of overlooking the government as a synergistic function in educational cooperation research. In practice, this paper offers specific counseling methods for encouraging the three parties to participate in educational cooperation, as well as establishing a mutually advantageous and win-win situation in educational cooperation.

6. The main conclusion of this manuscript appears in the third point of 6.1.1 Theoretical contribution, and the management implications appear in 6.1.2 Practical contribution section. Can these two parts form a one-to-one correspondence? Otherwise, the management implications are not deep-going. Reply:

Your question is very good, I have reorganized the contents of 6.1.1 "Theoretical Contribution" and 6.1.2 "Practical Contribution" to make the parts correspond to each other, I hope you will be satisfied with my modification.

Modify:

(The 6.1 Contributions of the study)

6.1.1 Theoretical contributions

The theoretical contribution of this work lies in the new findings and the construction of a new research model. On the one hand, it highlights the differences in the willingness of government, business, and universities to participate in educational collaborations and corrects the traditional view that government is most likely to dominate educational collaborations. Previous studies usually assume that the government should have the strongest willingness to participate, as enterprises and universities only participate in GEUS under the leadership of the government [60] However, this research comes to a different conclusion. In this paper, it is clear that the universities have the greatest desire to participate in GEUS, followed by the enterprises and then the government. The government is in a good position to act as a motivator and facilitator when both universities and enterprises want to participate.

On the other hand, this paper constructs a symbiotic evolutionary game model of government-enterprise-academia symbiosis that integrates the indicators of fusion revenue distribution ratio, which complements the indicator system of traditional research on government participation behavior. The current GEUS studies view the government as both an encourager and a punisher [16]. Therefore, only the government's function as an inducement and the punishment are taken into account. However, this article argues that the government will also play the role of the symbiont. If GEUS is successful, the government will allocate additional revenue to enterprises and universities, thus promoting the three together to make the "cake" of GEUS bigger. There is no longer a simple zero-sum game between the three parties, but win-win cooperation. Therefore, this study introduces the benefit allocation ratio. It is proved that GEUS can indeed be promoted by adjusting the benefits allocation ratio, and the larger the enterprise's share in the allocation ratio, the faster the three parties will choose GEUS.

6.1.2 Practical contribution

The practical contribution of this paper is to give specific strategies to promote better participation of government, enterprises, and universities in educational cooperation On the one hand, this paper gives strategies to accelerate educational cooperation in terms of the strength of the willingness of the three parties to participate in it, showing that active participation of the three parties can accelerate the evolution of the system, while passive participation can achieve the opposite effect. When the initial willingness of government and enterprises to participate changes, it does not affect the participation strategies of universities, but the effect between the two is particularly dramatic. Specifically, enterprises are the most inclined to a wait-and-see attitude, while the government is the most likely to opt not to participate. Furthermore, if the government's initial readiness to cooperate wanes, the government will first analyze enterprises' willingness to join. Only if enterprises are more willing to engage than before will the government opt to participate. Finally, the deeper the desire of enterprises to participate adversely, the greater the likelihood that the government will opt not to participate.

On the other hand, this paper provides three specific ways to accelerate educational cooperation in terms of incentives, penalties, and distribution mechanisms, which provide a grounded program for participants to better engage in educational cooperation. First, GEUS is unaffected by incentive allocation ratios. The government should focus on how to enhance the enterprise's benefit distribution ratio rather than focusing on how to set the incentive ratio. Because enterprises seek to maximize their economic benefits, changes in earnings distribution will trigger quick changes in their desire to participate. By contrast, universities are more concerned with academic values and social interests, along with talent development, scientific research, and social services [15]. Second, higher default fees should be imposed to expedite the GEUS process, too low default fees will eliminate the incentive for enterprises and governments to participate. The government can employ negative incentives to lead enterprises and universities to participate in GEUS by appropriately increasing the penalty for default. Third, increasing government revenues while decreasing government and enterprise expenditure can hasten GEUS, whereas cutting university expenditure has little influence on GEUS. Therefore, the government should establish a municipal GEUS fund to share risk with businesses, decreasing costs on both sides. Implement tax advantages for enterprises who are willing to actively engage, cutting their costs.

15. Orazbayeva B, Davey T, Plewa C, Galán-Muros V. Engagement of academics in education-driven university-business cooperation: a motivation-based perspective. Studies in Higher Education. 2020;45: 1723–1736. doi:10.1080/03075079.2019.1582013

16. Zan A, Yao Y, Chen H. University–industry collaborative innovation evolutionary game and simulation research: The agent coupling and group size view. IEEE Transactions on Engineering Management. 2019;68: 1406–1417. doi:10.1109/TEM.2019.2908206

60. Gustafsson R, Jarvenpaa S. Extending community management to industry‐university‐government organizations. R&d Management. 2018;48: 121–135. doi:10.1111/radm.12255

7. This manuscript needs careful editing by someone with expertise in technical English editing. Reply:

Your suggestions are very meaningful and I appreciate your tolerance during the review process. To improve the linguistic quality of the text, I have invited native speakers to polish the vocabulary and grammar of this paper, and I hope you are satisfied with the result of the polishing.

---

## [Decision Letter · Decision Letter 1]

8 Nov 2023

Educational Cooperation in the Perspective of Tripartite Evolutionary Game among Government, Enterprises and Universities

PONE-D-23-23142R1

Dear Dr. Zhang,

We’re pleased to inform you that your manuscript has been judged scientifically suitable for publication and will be formally accepted for publication once it meets all outstanding technical requirements.

Kind regards,

Tinggui Chen

Academic Editor

PLOS ONE

Additional Editor Comments (optional):

Reviewers' comments:

Reviewer's Responses to Questions

**Comments to the Author**

1. If the authors have adequately addressed your comments raised in a previous round of review and you feel that this manuscript is now acceptable for publication, you may indicate that here to bypass the “Comments to the Author” section, enter your conflict of interest statement in the “Confidential to Editor” section, and submit your "Accept" recommendation.

Reviewer #1: (No Response)

Reviewer #2: All comments have been addressed

2. Is the manuscript technically sound, and do the data support the conclusions?

Reviewer #1: (No Response)

Reviewer #2: No

3. Has the statistical analysis been performed appropriately and rigorously? 

Reviewer #1: (No Response)

Reviewer #2: N/A

4. Have the authors made all data underlying the findings in their manuscript fully available?

Reviewer #1: (No Response)

Reviewer #2: No

5. Is the manuscript presented in an intelligible fashion and written in standard English?

Reviewer #1: (No Response)

Reviewer #2: Yes

6. Review Comments to the Author

Reviewer #1: I am satisfied with the revisions carried out based on earlier feedback. The paper is in need of a final language check, preferably by an experienced or professional proofreader, to improve the clarity of expression and impact of your ideas. Once this is resolved, your paper will be ready for acceptance.

Reviewer #2: The modification of this paper is satisfactory. The authors are able to respond well to all comments.

7. PLOS authors have the option to publish the peer review history of their article (what does this mean?). If published, this will include your full peer review and any attached files.

Reviewer #1: No

Reviewer #2: No

---

## [Editor Report · Acceptance letter]

13 Dec 2023

PONE-D-23-23142R1 

PLOS ONE

Dear Dr. Zhang, 

I'm pleased to inform you that your manuscript has been deemed suitable for publication in PLOS ONE. Congratulations! Your manuscript is now being handed over to our production team.

Kind regards, 

on behalf of

Dr. Tinggui Chen 

Academic Editor

PLOS ONE